# Unsupervised Adaptation from Repeated Traversals for Autonomous Driving

**Yurong You**[*†1]   **Cheng Perng Phoo**[*1]   **Katie Z Luo**[*1]   **Travis Zhang**[1]
**Wei-Lun Chao**[2]   **Bharath Hariharan**[1]   **Mark Campbell**[1]   **Kilian Q. Weinberger**[1]
[1]Cornell University, Ithaca NY   [2]The Ohio State University, Columbus, OH

## Abstract

For a self-driving car to operate reliably, its perceptual system must generalize to the end-user's environment — ideally without additional annotation efforts. One potential solution is to leverage unlabeled data (e.g., unlabeled LiDAR point clouds) collected from the end-users' environments (i.e. target domain) to adapt the system to the difference between training and testing environments. While extensive research has been done on such an unsupervised domain adaptation problem, one fundamental problem lingers: there is no reliable signal in the target domain to supervise the adaptation process. To overcome this issue we observe that it is easy to collect unsupervised data from multiple traversals of repeated routes. While different from conventional unsupervised domain adaptation, this assumption is extremely realistic since many drivers share the same roads. We show that this simple additional assumption is sufficient to obtain a potent signal that allows us to perform iterative self-training of 3D object detectors on the target domain. Concretely, we generate pseudo-labels with the out-of-domain detector but reduce false positives by removing detections of supposedly mobile objects that are persistent across traversals. Further, we reduce false negatives by encouraging predictions in regions that are not persistent. We experiment with our approach on two large-scale driving datasets and show remarkable improvement in 3D object detection of cars, pedestrians, and cyclists, bringing us a step closer to generalizable autonomous driving. Code is available at https://github.com/YurongYou/Rote-DA.

## 1 Introduction

Autonomous vehicles and driver-assist systems require 3D object detectors to accurately identify and locate other traffic participants (cars, pedestrians and so on) to drive safely [26, 31, 12, 25, 21]. Modern 3D object detectors achieve high accuracy on benchmark datasets [9, 11, 41, 35, 36, 33, 42]. However, most benchmark data sets train and test classifiers on essentially the same locations (city, country), time, and weather conditions, and therefore represent the "best case" of an end-user using the self-driving car in precisely the same conditions it was trained on. A more realistic scenario is that self-driving cars trained in, for example, Germany will be driven in the USA. Unfortunately, past work has shown that this domain gap results in a catastrophic drop in accuracy [32]. Given that an end-user may choose to operate their car wherever they please, adapting the perception pipeline effectively to such domain shifts is a critical challenge.

An obvious solution is to retrain the detector in the target domain (i.e., the end-user's location/environment). Unfortunately, this requires large amounts of labeled data, where expert human

---

[*]Denotes equal contribution.
[†]Correspondences could be directed to yy785@cornell.edu

36th Conference on Neural Information Processing Systems (NeurIPS 2022).

annotators painstakingly locate every object in LiDAR scans in every conceivable location. Such labeled data is all but impossible to obtain in sufficient quantity. However, *unlabeled* data is not. No matter where the end-user intends to use their car, likely thousands of cars drive there every day already. By simply logging the data collected by cars with adequate drive-assist sensors, one obtains a wealth of information about the local environment, which should be useful to adapt a detector to this new domain. But it is unclear how to use this data: in particular, how can the detector learn to correct its many mistakes in this new domain if it has no labels at all?

The key here is the fact that this unlabeled data is not just an arbitrary collection of unrelated scenes. If we look at a population of cars driving around in a city, we observe that they all visit a shared set of roads and intersections. Indeed, as pointed out by [40], any single vehicle will probably be driven on the same route, day in and day out (e.g., commute, grocery shopping, patrol routes). Even when one end-user takes their car on a new route, it is likely that other cars have taken that very route not long before. This fact implies that the unlabeled data obtained from cars will typically contain *multiple traversals of the same route*, obtained for free without any targeted data collection. Previous work has already shown that aggregating data from multiple traversals can aid visual odometry [2] and unsupervised object discovery [40].

In this paper we argue that multiple traversals are particularly suited for end-user domain adaptation. We assume the existence of unlabeled LiDAR data from several repeated traversals of routes within the target domain (e.g. collected a few hours or days apart). For a LiDAR point captured in any one of these traversals, we use the other traversals to compute a persistency prior (PP-score) [40], capturing how persistent this LiDAR point has been across traversals: persistent points are likely static background. The PP-score thus yields a proxy signal for foreground vs background. This provides a powerful signal to correct both false positives and false negatives: detector outputs that mostly capture background points are likely false positives, and foreground points that are not captured by any detection reveal false negatives. To formalize this intuition, we propose a new iterative fine-tuning approach. We use the detector to generate 3D bounding boxes along the recorded traversals but remove boxes with lots of persistent (and thus static) points as false positives. We fine-tune the detector on this filtered data, and then "rinse and repeat". To reduce false negatives during this training, we introduce a new auxiliary loss that forces the detector to classify non-persistent LiDAR points as foreground. We refer to our method as *Rote Domain Adaptation (Rote-DA)*.

The resulting approach is a simple modification of existing object detectors, but offers substantial accuracy gains in unsupervised domain adaptation. We demonstrate on the Lyft [11] and Ithaca-356 [5] benchmark data sets that our approach consistently leads to drastic improvements when adapting a detector trained on KITTI [9] to the local environments — in some categories even outperforming a dedicated model trained on hand labeled data in the target domain (which we intended as an upper bound).

## 2 Related Works

We seek to adapt a 3D object detector from a source to a target domain with the help of unlabeled target data of *repeated traversals*.

**Unsupervised Domain Adaptation in 3D.** Improving generalizability of visual recognition systems (trained on a source domain) without annotated data from the testing environment (target domain) falls under the purview of unsupervised domain adaptation (UDA). The key to successful adaptation is leveraging the right information about the target domain. After all, without any knowledge of the target domain, adapting any learning systems would be extremely challenging, if not impossible. The most common source of information used for adaptation in the literature is the unlabeled data from the target domain; ST3D [37] improves conventional self-training adaption using stronger data augmentations and maintaining a memory bank of high quality predictions for self-training throughout adaptation; inspired by success in 2D UDA approaches that leverage feature alignment techniques [16, 29, 10, 20, 23], MLC-Net [17] proposes to encourage domain alignment by imposing consistency between a source detector and its exponential moving average [28] at point, instance, and neural-statistics-level on the target unlabeled data.

Other than unlabeled data, other work has also sought to use other information from the target domain to improve adaptation. One notable work along these lines is *statistical normalization* [32] where the authors identify car size difference as the biggest source of domain gap and propose to scale

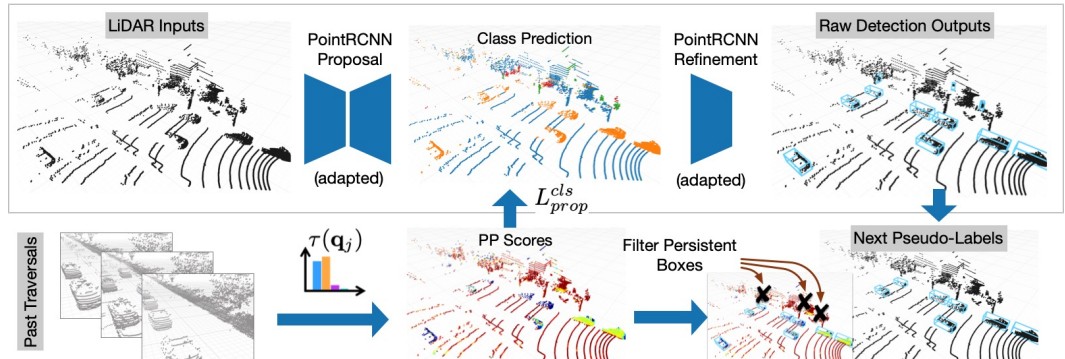

Figure 1: A schematic layout of Rote-DA. The PointRCNN Proposal network classifies each input LiDAR point as car/pedestrian/cyclist (Class Predictions) with three binary classifiers. The PP-Score is used during fine-tuning in an auxiliary loss function $L_{prop}^{cls}$ to reduce false-negatives. The Refinement network produces bounding boxes for the target data (Raw Detection Outputs). For the next self-training round, these are filtered with posterior and foreground/background filtering to reduce false positives, giving rise to the next pseudo-labels (bottom right).

the source data with the target data car size for adaptation. Knowing that the difference in weather conditions between source and target domain would cause changes in point cloud distributions, SPG [36] seeks to fill in points at foreground regions to address the domain gap. In addition to unlabeled data, other work [24, 38] has also explored temporal consistency — or more precisely, tracking of rigid 3D objects — to improve adaptation. Our work explores another rich yet easily attainable source of information — repeated traversals. In principle, one could combine our approach with prior UDA methods that use only the unlabeled data [37, 17] for additional marginal gains at the cost of increased algorithmic complexity. We choose to keep our contribution simple and clear and focus only on self-training with repeated traversals, which in itself is very effective and straightforward to replicate.

**Repeated Traversals.** Repeated traversals contain rich information that have already been used in a variety of scenarios. Early works utilize multiple traversals of the same route for localization [2, 14]. Repeated traversals of the same location allows discovering of non-stationary points in a point cloud captured by modern self-driving sensor since non-stationary points are less likely to persist across different traversals of the same location. To formalize this intuition, [2] develop an entropy-based measure, termed ephemerality score (see background 3.1) to determine dynamic points in a scene and subsequently, uses the signal to learn a representation for 2D visual odometry in a self-supervised manner. Building upon ephemerality, [40] utilize multiple common sense rules to discover a set of mobile objects for self-training a mobile object detector without any human supervision. Similar to [40] we leverage information from repeated traversals and use self-training; however in contrast to our work they focus on single class object discovery and our work is the first to show how multiple traversals can be used for domain adaptation. In addition to detecting foreground points/objects, repeated traversals have also been utilized by Hindsight [39] to decorate 3D point clouds with learned features for better 3D object detection. In principle, we could combine our approach with Hindsight to bring forth better generalizability but we did not explore such combination for simplicity and leave the exploration for future work.

## 3   Rote Domain Adaptation (Rote-DA)

We seek to adapt a 3D object detector pretrained on a certain area/domain (e.g. KITTI [8] in Germany) for reliable deployment to a different target area/domain (e.g., Lyft [11] in the USA). Without loss of generality, we assume all objects of interest are *dynamic* (e.g., cars, pedestrians, and cyclists). Similar to prior work [37, 36, 17, 32], we assume access to unlabeled target data for adaptation. Crucially different from previous work, we assume that the unlabeled target data are collected from the same routes repeatedly, and the localization information is available for adaptation. We note that such an additional assumption is highly realistic, since with current localization technology [4, 39] these data could be easily collected by the end-users going about their daily lives. For simplicity, we will focus

on adapting point-based detectors [30, 38, 39, 40, 18, 24], specifically PointRCNN [26] which is one of the current state-of-the-art 3D object detectors.

## 3.1 Background

Our work leverages persistence prior score (PP-score) from multiple traversals [40] to adapt PointR-CNN to a new, target domain. We review key concepts relevant to the understanding of our approach.

**Persistence prior score (PP-score) from multiple traversals.** The PP-score [40, 2] is an entropy-based measure that quantifies how persistent a single LiDAR point is across multiple traversals. We assume access to unlabeled LiDAR data that are collected from multiple traversals of a set of location $L$; each traversal contains a series LiDAR scans. To calculate the PP-score, we further assume that these LiDAR scans of a traversal $t$ have been pre-processed, such that the LiDAR points around a location $g \in L$ are aggregated to form a dense point cloud $\boldsymbol{S}_g^t$. We note that $\boldsymbol{S}_g^t$ is only used for PP-score computation, not as an input to 3D object detectors.

Given a single 3D point $\boldsymbol{q}$ around location $g$, we can calculate its PP-score by the following steps. First, we count the number of its neighboring points within a certain radius $r$ (say 0.3m) in each $\boldsymbol{S}_g^t$:

$$N_t(\boldsymbol{q}) = \left|\{\boldsymbol{p}_i \mid \|\boldsymbol{p}_i - \boldsymbol{q}\|_2 < r, \boldsymbol{p}_i \in \boldsymbol{S}_g^t\}\right|. \tag{1}$$

We then normalize $N_t(\boldsymbol{q})$ across traversals $t \in \{1, \cdots, T\}$ into a categorical probability:

$$P(t; \boldsymbol{q}) = \frac{N_t(\boldsymbol{q})}{\sum_{t'=1}^T N_{t'}(\boldsymbol{q})}. \tag{2}$$

With $P(t; \boldsymbol{q})$, we can then compute the PP-score $\tau(\boldsymbol{q})$ by

$$\tau(\boldsymbol{q}) = \begin{cases} 0 & \text{if } N_t(\boldsymbol{q}) = 0 \ \ \forall t; \\ \frac{H(P(t;\boldsymbol{q}))}{\log(T)} & \text{otherwise}, \end{cases} \tag{3}$$

where $H$ is the information entropy. Essentially, the more uniform $P(t; \boldsymbol{q})$ is across traversals, the higher the PP-score is. This happens when the neighborhood of $\boldsymbol{q}$ is stationary across traversals; i.e., $\boldsymbol{q}$ is likely a background point. In contrast, a low PP-score indicates that some traversals $t$ have much higher $P(t; \boldsymbol{q})$ than some other traversals. This suggests that the neighborhood of $\boldsymbol{q}$ is sometimes empty (so low probability) and sometimes occupied (e.g., by a foreground car, so high probability), and when $\boldsymbol{q}$ is detected by LiDAR, it is likely reflected from a foreground object.

**PointRCNN.** PointRCNN [26] is a two-stage detector. In the first stage, each LiDAR point is classified into a foreground class or background, and a 3D box proposal is generated around each foreground point. The proposals are then passed along to the second stage for bounding box refinement, which refines both the class label and box pose. It is worth noting that this two-stage pipeline is widely adopted in many other detectors [6, 22, 34, 1]. An understanding and solution to the error patterns of PointRCNN, especially when it is applied to new environments, are thus very much applicable to other detectors.

By taking a deeper look at the inner working of PointRCNN, we found that if a foreground LiDAR point is misclassified as the background in the first stage, then it is removed from consideration for the refinement (i.e., bound to be a false negative). In our approach, we thus propose to incorporate the PP-score to correct this error during iterative fine-tuning. In the following, we describe the original loss function used to train PointRCNN's first stage.

Let us denote by $N_c$ the number of foreground classes and by $N_p$ the number of points in a scene. An annotated point cloud can be represented by a set of tuples $\{(\boldsymbol{q_i}, \boldsymbol{y_i}, \boldsymbol{b_i})\}_{i=1}^{N_p}$, where $\boldsymbol{y_i}$ is a one-hot $N_c$-dimensional class label vector and $\boldsymbol{b_i}$ is the bounding box pose that encapsulates $\boldsymbol{q_i}$. The loss function can be decomposed into two terms:

$$L(\{\boldsymbol{q_i}, \boldsymbol{y_i}, \boldsymbol{b_i}\}_{i=1}^{N_p}) = \sum_{i=1}^{N_p} L_{\text{cls}}(\boldsymbol{q_i}, \boldsymbol{y_i}) + L_{\text{reg}}(\boldsymbol{q_i}, \boldsymbol{b_i}). \tag{4}$$

The first term $L_{\text{cls}}$ is for per-point classification (or equivalently, *segmentation* of the point cloud). The second term $L_{\text{reg}}$ is for proposal regression. For the former, a focal loss [15] is used:

$$\frac{1}{\alpha} L_{\text{cls}}(\boldsymbol{q_i}, \boldsymbol{y_i}) = \sum_{c=1}^{N_c} y_{ic} (1 - p_c)^\gamma \log(p_c) + (1 - y_{ic})(p_c)^\gamma \log(1 - p_c) \tag{5}$$

where $p_c$ is the one-vs-all probability of class $c$, produced by PointRCNN's first stage; $y_{ic}$ indexes the $c$-th position of $\boldsymbol{y_i}$; $\alpha$ and $\gamma$ are hyperparameters for the focal loss (we use default value $\alpha = 0.25$ and $\gamma = 2.0$).

## 3.2 Adaptation Strategy

**Approach overview.** Our adaptation approach is built upon the conceptually-simple but highly effective self-training for adaptation [13, 38]. The core idea is to iteratively apply the current model to obtain *pseudo-labels* on the unlabeled target data, and use the pseudo-labels to fine-tune the current model. The current model is initialized by the source model (i.e., a pre-trained PointRCNN). Self-training works when the pseudo-labels are of high quality — in the ideal case that the pseudo-labels are exactly the ground truths, self-training is equivalent to supervised fine-tuning. In practice, the pseudo-labels can be refined through the iterative process, but errors may also get reinforced. It is therefore important to have a "quality control" mechanism on pseudo-labels.

In this section, we propose a set of novel approaches to improve the quality of pseudo-labels, taking advantage of the PP-scores, illustrated in Figure 1.

**Pseudo-label refinement for false positives removal.** Pseudo-labels generated by the source model are often noisy when the source model is first applied to the target data that are different from the source data. As shown in [32] and our experiments, PointRCNN suffers a serious performance drop in new environments. Many of the detected boxes are false positives or false negatives. To control the quality, we first leverage the PP-score to identify and filter out false positives.

To assess the quality of a bounding box $b$, we first crop out the points $\{\boldsymbol{q_j}\}_{j \in b}$ in it, and query their PP-scores $\{\tau(\boldsymbol{q_j})\}_{j \in b}$. A bounding box is highly likely to be a false positive if it contains so many *persistent* points, i.e., points with high PP-scores. To this end, we summarize the PP-scores $\{\tau(\boldsymbol{q_j})\}_{j \in b}$ of a box by the $\alpha_{\text{FB-F}}$ percentile, and remove the box if the value is larger than a threshold $\gamma_{\text{FB-F}}$. In our experiments, we set $\alpha_{\text{FB-F}} = 20$ and $\gamma_{\text{FB-F}} = 0.5$ (We find these values are not sensitive). Since we are effectively filtering out boxes that do not respect the foreground/background segmentation obtained from multiple traversals, we term this filtering approach *Foreground Background Filtering*(**FB-F**).

In addition, we present another complementary way to identify another kind of false positives. Essentially, FB-F can effectively identify false positives that should have been detected as background. However, it cannot identify false positives that result from wrong classification or size estimates. Indeed, after FB-F, we still see a decent amount of this kind of false positives; the remaining pseudo-labels are more numerous than the ground-truth boxes. One naive way to remove them is by thresholding the model's confidence on them. However, setting a suitable threshold is nontrivial in self-training, since the model's confidence will get higher along the iterations.

We thus propose to directly set a cap on the average number of pseudo-labels per class $c$ in a scene. We make the following assumption: as long as the source and target domains are not from drastically different areas (e.g., a city vs. barren land), the object frequency in the source domain can serve as a good indicator for what a well-performing object detector should see in the target domain. To this end, we set the cap by $\beta \times \frac{N_c^{\mathcal{S}}}{N_{\text{scenes}}^{\mathcal{S}}}$, where $N_{\text{scenes}}^{\mathcal{S}}$ and $N_c^{\mathcal{S}}$ are the total number of source training scenes and the ground-truth objects of class $c$ in them, respectively. The value $\beta \in [0, 1]$ is a hyperparameter that controls the tightness of the cap. With this cap, after creating pseudo-labels on $N_{\text{scenes}}^{\mathcal{T}}$ target scenes we keep the top $\beta \times \frac{N_c^{\mathcal{S}}}{N_{\text{scenes}}^{\mathcal{S}}} \times N_{\text{scenes}}^{\mathcal{T}}$ of them for each class $c$ according to the model's confidence. Given we control the distribution of objects (similar to posterior regularization [7]), we term this filtering step Posterior Filtering (PO-F).

**Foreground Background Supervision (FB-S) for false negatives reduction.** FB-F, as discussed above, can effectively filter out false positives that should have been background. Now we show

that the PP-scores are also useful for correcting false negatives. As mentioned in subsection 3.1, the first stage of PointRCNN is the key to false negatives: if a foreground point is misclassified as background, then it is bound to be a false negative. To rectify this, we incorporate the PP-score into the fine-tuning process. Specifically, we modify the pseudo-class-label $\hat{y}_i$ of a point $q_i$ in Equation 5 with PP-score $\tau(q_i)$:

$$y_i = \begin{cases} \mathbf{0} & \text{if } \tau(q_i) > \tau_U, \\ \mathbf{1} & \text{if } \tau(q_i) < \tau_L \text{ and } \hat{y}_i = \mathbf{0}, \\ y_i & \text{otherwise.} \end{cases} \tag{6}$$

where $\mathbf{0}$ is a zero vector and $\mathbf{1}$ is an all-one vector. Essentially, if a point is persistent (i.e., high $\tau(q_i)$), we set the the pseudo-class-label as background $\mathbf{0}$. On the contrary, for a non-persistent point (i.e., low $\tau(q_i)$) that is deemed as background (i.e., $\hat{y}_i = \mathbf{0}$) by the current model, we encourage the scores of all the foreground classes to be as high as possible, so that a foreground proposal can be generated. We note that while this foreground class label may be wrong, the subsequent refinement by PointRCNN's second stage can effectively correct it.

## 4  Experiments

**Datasets.** We validate our approach on a single source dataset, the KITTI dataset [8] and two target datasets: the Lyft Level 5 Perception dataset [11] and the Ithaca-365 dataset [5]. The KITTI dataset is collected in Karlsruhe, Germany, while the Lyft and Ithaca-365 dataset is collected in Palo Alto (California) and Ithaca (New York) in the US respectively. Such setup is chosen to simulate large domain difference [32]. To show good generalizability, we use *exactly the same* hyper-parameters for adaptation experiments on these two target datasets.

To the best of our knowledge[3], Lyft and Ithaca-365 are the only two publicly available autonomous driving datasets that have both bounding box annotations and multiple traversals with accurate 6-DoF localization. We use these two datasets to test out two different adaptation scenarios. The first scenario is that the detector is trained on data from nearby locations, *but not from the roads and intersections it will be driven on*. Thus, following [40], we split the Lyft dataset so that the "train"/test set are *geographically disjoint*; we also discard locations with less than 2 traversals in the "train" set. This results a "train"/test split of 11,873/4,901 point clouds for the Lyft dataset. We use all traversals available (2-10 in the dataset) to compute PP-score for each scene.

The second adaptation scenario is when the detector uses unlabeled data *from the same routes that it sees at test time*. This scenario is highly likely in practice since, as mentioned before, a self-driving car can leverage data collected by other cars on the same route previously. To test this, we split the Ithaca-365 dataset based on the data collection date, keeping the same geographical locations in both train and test. This results in 4445/1644 point clouds. The "train" sets of these two target datasets are used without labels. We use the roof LiDAR (40/60-beam in Lyft; 128-beam in Ithaca-365), and the global 6-DoF localization with the calibration matrices directly from the raw data. We do not use the intensity channel of the LiDAR data due to drastic difference in sensor setups between datasets. We use 5 traversals to compute PP-score for each scene.

We pre-train the 3D object detection models on the train split (3,712 point clouds) of KITTI datasets to detect *Car*, *Pedestrian* and *Cyclist* classes, and adapt them to detect the same objects in the Lyft and *Car*, *Pedestrian* in the Ithaca-365 since there too few Cyclist in the dataset to provide reasonable performance estimate. Since KITTI only provides 3D object labels within frontal view, we focus on frontal view object detection only during adaption and evaluation.

**Evaluation metric.** On the Lyft dataset, we follow [39] to evaluate object detection in the bird's-eye view (BEV) and in 3D for the mobile objects by KITTI [9] metrics and conventions: we report average precision (AP) with the intersection over union (IoU) thresholds at 0.7/0.5 for Car and 0.5/0.25 for Pedestrian and Cyclist. We further follow [32] to evaluate the AP at various depth ranges. Due space constraint, we present $AP_{BEV}$ at IoU=0.7 for Car and 0.5 for Pedestrian and Cyclist in the main text and defer the rest of the results to the supplementary materials. On the Ithaca-365 dataset, the default match criterion is by the minimum distance to ground-truth bounding boxes. We evaluate

---

[3]We note that though there are some scenes with multiple traversals in the nuScenes dataset [3] as used in [39, 40], the localization in $z$-axis is not accurate (https://www.nuscenes.org/nuscenes#data-format).

Table 1: **Detection performance of KITTI → Lyft adaptation.** Given a PointRCNN detector [26] pre-trained on the KITTI dataset, adaptation strategies improves its detection performance on the target Lyft dataset. We breakdown their detection $AP_{BEV}$ by depth ranges. We also show in-domain performance of the same model (training and testing on the Lyft dataset) as a reference. Please refer to supplementary material for corresponding $AP_{3D}$ results and results under other IoU metrics, where we observe a similar trend. * ST3D's adaptation involves 30 epochs of self-training by defaults so for fair comparison, we show ST3D's results early-stopped at the 10-th epoch.

| Method | Car | | | | Pedestrian | | | | Cyclist | | | |
|---|---|---|---|---|---|---|---|---|---|---|---|---|
| | 0-30 | 30-50 | 50-80 | 0-80 | 0-30 | 30-50 | 50-80 | 0-80 | 0-30 | 30-50 | 50-80 | 0-80 |
| No Adaptation | 57.1 | 33.9 | 9.0 | 35.4 | 37.1 | 21.6 | 0.6 | 19.1 | 43.7 | 8.2 | 0.0 | 24.4 |
| ST3D (R10)* | 64.8 | 52.7 | 27.7 | 49.2 | 35.6 | 25.6 | 0.0 | 19.3 | 56.9 | 15.0 | 0.0 | 33.3 |
| ST3D (R30) | 66.4 | 52.5 | **29.1** | 50.2 | 0.0 | 0.0 | 0.0 | 0.0 | 11.9 | 2.5 | 0.0 | 7.1 |
| Rote-DA (Ours) | **69.0** | **58.8** | 22.6 | **52.1** | **48.1** | **40.8** | 2.6 | **28.7** | **64.7** | **26.4** | 0.0 | **40.0** |
| SN | 75.4 | 59.9 | 23.2 | 54.5 | 41.4 | 32.8 | 1.8 | 24.2 | 39.7 | 8.7 | 0.0 | 22.4 |
| In Domain | 75.9 | 69.0 | 43.5 | 64.3 | 43.0 | 38.4 | 11.2 | 31.6 | 65.0 | 31.4 | 0.7 | 41.6 |

Table 2: **Detection performance of KITTI → Ithaca-365 adaptation.** We evaluate the mAP as described in section 4 by different depth ranges and object types. Please refer to Table 1 for namings.

| Method | Car | | | | Pedestrian | | | |
|---|---|---|---|---|---|---|---|---|
| | 0-30 | 30-50 | 50-80 | 0-80 | 0-30 | 30-50 | 50-80 | 0-80 |
| No Adaptation | 54.3 | 32.1 | 1.2 | 29.3 | 52.1 | 21.1 | 0.0 | 25.4 |
| ST3D (R10) | 66.2 | 38.8 | 3.8 | 36.3 | 53.0 | 23.9 | 0.0 | 26.5 |
| ST3D (R30) | 65.4 | 28.5 | 9.9 | 33.3 | 47.9 | 24.9 | 0.0 | 25.7 |
| Rote-DA (Ours) | **66.9** | **43.5** | **15.6** | **43.5** | **53.6** | **33.0** | **0.2** | **31.2** |
| SN | 54.7 | 33.0 | 2.0 | 30.0 | 52.3 | 22.2 | 0.0 | 26.0 |
| In Domain | 72.4 | 50.1 | 24.4 | 50.5 | 55.3 | 29.9 | 2.6 | 32.7 |

the mean of AP with match thresholds of {0.5, 1, 2, 4} meters for Car and Pedestrian. We follow [39] to evaluate only detection in frontal view.

**Implementation of PointRCNN.** We use the default implementation/configuration of PointR-CNN [26] from OpenPCDet [19]. For fine-tuning, we fine-tune the model for 10 epochs with learning rate $1.5 \times 10^{-3}$ (pseudo-labels are regenerated and refined after each epoch). All models are trained/fine-tuned with 4 GPUs (NVIDIA 2080Ti/3090/A6000).

**Comparisons.** We compare the proposed method against two methods with publicly available implementation: Statistical Normalization (SN) [32] and ST3D [37]. SN *assumes access to mean car sizes of target domain*, and applies object sizes scaling to address the domain gap brought by different car sizes. Since there is less variability on box sizes among pedestrians and cyclists, we only scale the car class using the target domain statistics. ST3D achieves adaptation via self-training on the target data with stronger augmentation and maintaining a memory bank of high quality pseudolabels.

### 4.1 Adaptation performance on KITTI → Lyft and Ithaca-365

In Table 1 and Table 2 we show the adaptation performance of adapting a KITTI pre-trained PointRCNN detection model to the Lyft and the Ithaca-365 datasets. We observe that despite its simplicity, Rote-DA outperforms all baselines on almost all metrics, across both datasets and across object types, confirming the potent learning signal from multiple traversals. Note that the hyper-parameters are kept as exactly the same between experiments in these two datasets, showing the strong generalizability of Rote-DA. While SN is more accurate than Rote-DA for cars on Lyft, it uses external information about car sizes that is unavailable to the other techniques, and that is not useful for other classes.

Table 3: **Ablation study on different components in Rote-DA.** Different from vanilla self-training, Rote-DA includes two additional components: pseudolabels refinement and Foreground Background Supervision (FB-S). In particular, the pseudo-labels refinement can be further subdivided into two subcomponents: FB-F and PO-F. We show detection performance ($AP_{BEV}$) of variants of Rote-DA without either of these two parts. We report performance at R10 for all variants.

| PO-F | FB-F | FB-S | Car 0-30 | 30-50 | 50-80 | 0-80 | Pedestrian 0-30 | 30-50 | 50-80 | 0-80 | Cyclist 0-30 | 30-50 | 50-80 | 0-80 |
|---|---|---|---|---|---|---|---|---|---|---|---|---|---|---|
| | | | 58.3 | 38.3 | 14.0 | 38.3 | 22.7 | 12.4 | 0.2 | 10.7 | 27.6 | 0.2 | 0.0 | 9.4 |
| ✓ | | | 68.8 | 52.7 | 12.3 | 46.0 | 46.3 | 30.9 | 0.0 | 22.5 | **66.3** | 6.3 | 0.0 | 35.2 |
| | ✓ | ✓ | 61.2 | 40.1 | 14.6 | 40.9 | 41.4 | 29.7 | 0.8 | 23.4 | 47.5 | 2.7 | 0.0 | 23.3 |
| ✓ | | ✓ | 68.4 | 55.4 | 19.6 | 49.0 | 41.7 | 30.8 | 1.4 | 21.8 | 43.3 | 1.7 | 0.0 | 21.2 |
| ✓ | ✓ | | **71.6** | 54.9 | 19.1 | 50.4 | **52.0** | 38.1 | **4.2** | **29.4** | 58.3 | 19.6 | 0.0 | 34.8 |
| ✓ | ✓ | ✓ | 69.0 | **58.8** | **22.6** | **52.1** | 48.1 | **40.8** | 2.6 | 28.7 | 64.7 | **26.4** | 0.0 | **40.0** |
| No Adaptation | | | 57.1 | 33.9 | 9.0 | 35.4 | 37.1 | 21.6 | 0.6 | 19.1 | 43.7 | 8.2 | 0.0 | 24.4 |

Rote-DA works especially well on the challenging categories of pedestrians and cyclists, almost doubling the performance on cyclists and even outperforming an in-domain detector in some scenarios (pedestrians, 0-30 m range). In contrast, prior domain adaptation strategies actually *hurt* performance for these categories. For e.g., ST3D through the course of self-training gradually over-fits to cars and "forgets" pedestrians and cyclists (comparing row ST3D(R10) and ST3D(R30), see also Figure 4).

Interestingly, when Rote-DA has access to unlabeled data from past traversals of the test routes (as on the Ithaca-365 dataset), the performance gains are even more significant, especially on the mid-to-far ranges (30-80m), improving accuracy by more than $10\times$ for cars in the 50-80m range.

## 4.2 Analysis

Unless otherwise stated, we conduct the following study on the Lyft dataset.

**Effects of different components.** We ablate different components of Rote-DA: pseudo-labels refinement and Foreground Background Supervision (FB-S) in Table 3. To start, vanilla self-training without any of the components would only yield marginal improvements to detecting cars whereas performance of the adapted detectors for the rarer classes (pedestrians, cyclists) degrade significantly compared to no adaptation. Posterior Filtering (PO-F) is an effective strategy to prevent performance degradation. Combining Foreground Background Filtering (FB-F) with PO-F would always yield significant improvements regardless of classes, showing usefulness of using PP-score for filtering and the efficacy of our filtering pseudo-label refinement strategy. Combining the Foreground Background Supervision (FB-S) with only PO-F would not be effective always but combining FB-S with the full pseudo-label refinement procedure would would bring forth significant improvements especially on cyclist.

**Effects of different rounds of iterative fine-tuning.** As customary to any iterative approach, we analyze the effect of the number of rounds of self-training in Figure 2. One conclusion is immediate: vanilla self-training degrades (even underperforms no adaptation) over more rounds of self-training potentially due to learning from erroneous pseudo-labels. Rote-DA (and its variants) improves for the first few rounds of training (before the 10th round), and experience little to no performance degradation over more rounds of training.

**Effect of Foreground Background Supervision (FB-F).** FB-F seeks to reduce false negatives by correcting the foreground predictions by the model. To validate this claim, we plot the precision-recall curves of various detectors in Figure 3. Comparing Rote-DA and PO-F +FB-F, we observe that the max recall for Rote-DA is much higher than PO-F +FB-F, suggesting FB-F is encouraging the detector to produce more meaningful boxes at foreground regions, thus reducing false negatives.

**Qualitative visualization.** In Figure 4, we visualize the adaptation results of various adaptation strategies in both Lyft and Ithaca-365 datasets. We observe that, aligning with quantitative results, ST3D has a good coverage of cars but usually ignores pedestrians and cyclists and generates many

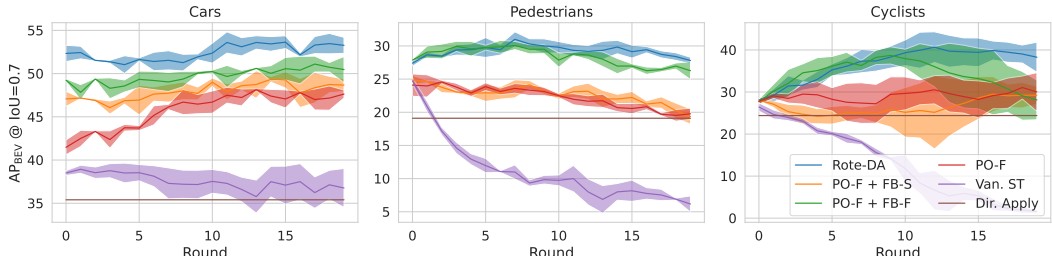

Figure 2: Performance of various detectors on KITTI → Lyft for different rounds of self-training (averaged across 3 runs with mean and one standard deviation reported). Van. ST stands for vanilla self-training without any modification; Dir. Apply stands for direct applying the source detector without any adaptation. We observe that the performance for vanilla self-training degrades over more rounds of self-training whereas variants of Rote-DA experience little to no degradation in performance after 10 rounds of training.

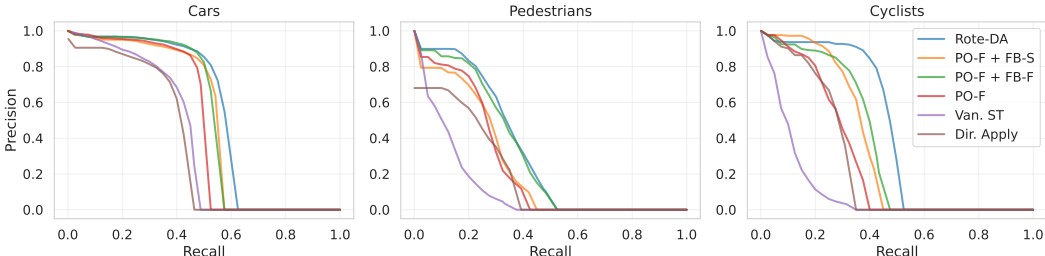

Figure 3: **Precision-recall curves on the KITTI → Lyft with 10 rounds of self-training.** We show the P-R curves of ablated Rote-DA, please refer to Figure 2 for naming. The precision and recall is calculated under $AP_{BEV}$ with IoU=0.7 for Cars, 0.5 for Pedestrians and Cyclists.

false positive cars; SN successfully corrects the car size bias, but can hardly improve the recall of the detection; Rote-DA adapts to the object size bias in the target domain while having a good recall rate of all three object classes.

**Additional results, analyses, qualitative visualization.** Please refer to the supplementary material for evaluation with more metrics, results on a different detection model (PVRCNN [25]) and on a different adaptation scenario (Waymo Open Dataset [27] to Ithaca-365), and more qualitative results.

## 5 Discussion

**Privacy concerns.** As our method relies on collecting unlabeled repeated traversal of the same routes, there are privacy concerns that have to be addressed before public deployment. This could be achieved by making data collection an opt-in option for drivers. Also, the collected data should be properly annoymized, or reduced to random road segments to remove any potential personal identifiable information.

**Limitations.** Our method currently focuses on adapting *dynamic*, i.e. mobile, object detectors to target domains using multiple traversals. However, Rote-DA could be extended to *static* objects easily via selecting the appropriate thresholds for Foreground Background Filtering and Foreground Background Supervision. We leave this exploration for future work. Also, we assume the source and target domain share the same object frequency for Posterior Filtering. However, this assumption could be alleviated via querying local authorities for the object frequency or by estimating the object frequencies from similar regions (we assume access to the locations of target domain).

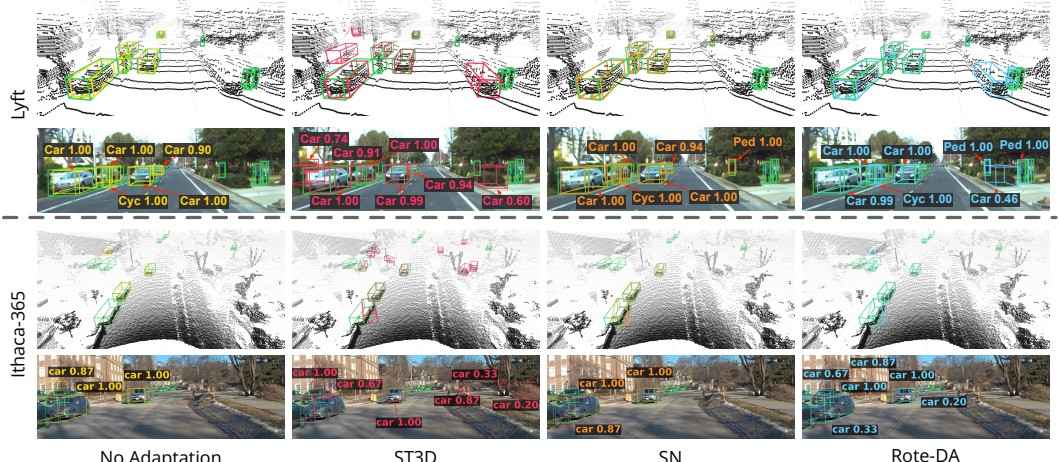

Figure 4: **Qualitative visualization of adaptation results.** We visualize one example scene (up:LiDAR, bottom: image (not used in the adaptation)) in the Lyft and Ithaca-365 test split, and plot the detection results with various adaptation strategies. Ground-truth bounding boxes are shown in green, detection boxes of no adaptation, ST3D, SN and Rote-DA are shown in yellow, red, orange and cyan, respectively. Zoom in for details. Best viewed in color.

## 6    Conclusion

End-user domain adaptation is one of the key challenges towards safe and reliable self-driving vehicles. In this paper we claim that unlike most domain adaptation settings in machine learning, the self-driving car setting naturally gives rise to a weak supervision signal that is exceptionally well-suited to adapt 3D object detector to a new environment. As drivers share roads, unlabeled LiDAR data automatically comes in the form of multiple traversals of the same routes. We show that with such data we can iteratively refine a detector to new domains. This is effective because we prevent it from reinforcing mistakes with three "safe guards": 1. Posterior Filtering, 2.Foreground Background Filtering, 3. Foreground Background Supervision. Although the experiments in this paper already indicate that Rote Domain Adaptation may currently be the most effective approach for adaptation in the self-driving context, we believe that the true potential of this method may be even greater than our paper seems to suggest. As cars with driver assist features become common place, collecting unlabeled data will become easier and cheaper. This could give rise to unlabeled data sets that are several orders of magnitudes larger than the original source data set, possibly yielding consistently more accurate detectors than are obtainable with purely hand-labeled training sets.

## 7    Acknowledgement

This research is supported by grants from the National Science Foundation NSF (IIS-1724282, TRIPODS-1740822, IIS-2107077, OAC-2118240, OAC-2112606 and IIS-2107161), the Office of Naval Research DOD (N00014-17-1-2175), the DARPA Learning with Less Labels program (HR001118S0044), the Cornell Center for Materials Research with funding from the NSF MRSEC program (DMR-1719875).

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
