# Supplementary Material for
# Unsupervised Adaptation from Repeated Traversals for Autonomous Driving

## S1 Implementation Details

The parameters that we used in this work were $\beta = 0.333$, and $N_c^{\mathcal{S}}$ values are 14357, 2207, and 734 for Cars, Pedestrians, and Cyclists, respectively. We include an ablation table for different values of $\beta$ in Table S1. For the focal loss, we set $\alpha = 0.25$ and $\gamma = 2.0$ which are the default values. For the Posterior Filtering, we set $\alpha_{\text{FB-F}} = 20$ and $\gamma_{\text{FB-F}} = 0.5$. We selected the best hyperparameters based on the performance on KITTI $\rightarrow$ Lyft and used the same hyperparameters for the rest of the settings.

Table S1: $\beta$-**Value Experiment Results.** Evaluated under $\text{AP}_{\text{BEV}}$ with IoU 0.7 for Car, 0.5 for Pedestrian and Cyclists. We show results experiementing with different $\beta$ parameters.

| | Car | | | | Pedestrian | | | | Cyclist | | | |
|---|---|---|---|---|---|---|---|---|---|---|---|---|
| $\beta$-Values | 0-30 | 30-50 | 50-80 | 0-80 | 0-30 | 30-50 | 50-80 | 0-80 | 0-30 | 30-50 | 50-80 | 0-80 |
| 0.333 | 69.0 | 58.8 | 22.6 | 52.1 | 48.1 | 40.8 | 2.6 | 28.7 | 64.7 | 26.4 | 0.0 | 40.0 |
| 0.500 | 65.5 | 61.4 | 26.4 | 53.4 | 39.2 | 31.3 | 1.2 | 20.4 | 52.0 | 20.3 | 0.0 | 33.0 |
| 0.666 | 61.9 | 51.9 | 19.1 | 48.3 | 40.8 | 29.0 | 1.0 | 20.4 | 46.5 | 14.6 | 0.0 | 28.3 |

## S2 Additional Detection Evaluation on the Lyft dataset

### S2.1 On different metrics.

We include additional evaluations on the Lyft dataset. In Tables S2, S3, and S4 we show the results with metrics $\text{AP}_{\text{3D}}$ at IoU 0.7 (cars) / 0.5 (pedestrian and cyclists), $\text{AP}_{\text{BEV}}$ at IoU 0.7 / 0.5, and $\text{AP}_{\text{3D}}$ at IoU 0.5 / 0.25, respectively. This corresponds to Table 1 in the main paper.

Table S2: **Detection performance of KITTI $\rightarrow$ Lyft adaptation.** Evaluated under $\text{AP}_{\text{3D}}$ with IoU 0.7 for Car, 0.5 for Pedestrian and Cyclist. Please refer to Table 1 for naming.

| | Car | | | | Pedestrian | | | | Cyclist | | | |
|---|---|---|---|---|---|---|---|---|---|---|---|---|
| Method | 0-30 | 30-50 | 50-80 | 0-80 | 0-30 | 30-50 | 50-80 | 0-80 | 0-30 | 30-50 | 50-80 | 0-80 |
| No Adaptation | 22.3 | 6.9 | 1.2 | 10.8 | 29.9 | 16.5 | 0.5 | 15.2 | 35.4 | 5.6 | 0.0 | 19.0 |
| ST3D (R10)* | 37.6 | 23.2 | 6.0 | 23.3 | 27.1 | 23.1 | 0.0 | 15.6 | 48.6 | 12.4 | 0.0 | 27.0 |
| ST3D (R30) | **44.1** | **26.8** | 5.0 | 26.5 | 0.0 | 0.0 | 0.0 | 0.0 | 10.7 | 2.5 | 0.0 | 6.3 |
| Rote-DA (Ours) | 43.5 | 25.9 | **7.8** | **27.4** | **36.3** | **35.2** | **2.6** | **23.0** | **57.7** | **22.1** | 0.0 | **35.0** |
| SN | 57.1 | 30.7 | 6.5 | 33.3 | 31.4 | 25.8 | 1.5 | 18.6 | 31.1 | 6.3 | 0.0 | 17.2 |
| In Domain | 63.5 | 43.1 | 15.9 | 43.1 | 34.6 | 30.1 | 8.3 | 24.6 | 59.4 | 25.2 | 0.4 | 36.2 |

Table S3: **Detection performance of KITTI $\rightarrow$ Lyft adaptation.** Evaluated under $\text{AP}_{\text{BEV}}$ with IoU 0.5 for Car, 0.25 for Pedestrian and Cyclist. Please refer to Table 1 for naming.

| | Car | | | | Pedestrian | | | | Cyclist | | | |
|---|---|---|---|---|---|---|---|---|---|---|---|---|
| Method | 0-30 | 30-50 | 50-80 | 0-80 | 0-30 | 30-50 | 50-80 | 0-80 | 0-30 | 30-50 | 50-80 | 0-80 |
| No Adaptation | 81.0 | **68.8** | 27.9 | 60.2 | 55.4 | 26.8 | 0.7 | 26.5 | **70.3** | 19.7 | 0.0 | 41.3 |
| ST3D (R10) | **82.2** | 68.3 | 36.3 | **64.0** | 48.1 | 27.3 | 0.0 | 24.0 | 69.2 | 23.6 | 0.0 | 41.6 |
| ST3D (R30) | 80.3 | 65.5 | **37.1** | 62.7 | 0.0 | 0.0 | 0.0 | 0.0 | 15.0 | 2.5 | 0.0 | 7.5 |
| Rote-DA (Ours) | 78.8 | 66.4 | 28.3 | 59.4 | **62.7** | **44.6** | **2.8** | **34.8** | 69.6 | **34.4** | 0.2 | **44.7** |
| SN | 81.3 | 65.5 | 30.7 | 60.1 | 53.9 | 38.5 | 2.0 | 30.2 | 67.3 | 26.9 | 2.5 | 42.6 |
| In Domain | 85.7 | 76.5 | 58.1 | 74.7 | 59.2 | 44.8 | 15.1 | 40.2 | 67.7 | 35.1 | 1.3 | 44.2 |

Table S4: **Detection performance of KITTI → Lyft adaptation.** Evaluated under $AP_{3D}$ with IoU 0.5 for Car, 0.25 for Pedestrian and Cyclist. Please refer to Table 1 for naming.

| Method | Car | | | | Pedestrian | | | | Cyclist | | | |
|---|---|---|---|---|---|---|---|---|---|---|---|---|
| | 0-30 | 30-50 | 50-80 | 0-80 | 0-30 | 30-50 | 50-80 | 0-80 | 0-30 | 30-50 | 50-80 | 0-80 |
| No Adaptation | 78.2 | 62.9 | 19.8 | 55.1 | 55.4 | 26.7 | 0.7 | 26.4 | **70.3** | 19.3 | 0.0 | 41.2 |
| ST3D (R10)* | **81.5** | **66.4** | 33.3 | **61.9** | 48.1 | 27.3 | 0.0 | 24.0 | 69.2 | 23.6 | 0.0 | 41.6 |
| ST3D (R30) | 79.7 | 64.5 | **33.5** | 60.9 | 0.0 | 0.0 | 0.0 | 0.0 | 15.0 | 2.5 | 0.0 | 7.5 |
| Rote-DA (Ours) | 76.4 | 65.4 | 25.6 | 57.0 | **62.2** | **44.5** | **2.8** | **34.6** | 69.6 | **34.4** | **0.2** | **44.1** |
| SN | 81.2 | 64.4 | 26.8 | 59.2 | 53.9 | 38.5 | 2.0 | 30.2 | 67.2 | 26.5 | 2.5 | 42.4 |
| In Domain | 83.8 | 74.4 | 51.7 | 72.0 | 59.2 | 44.5 | 14.8 | 40.1 | 67.7 | 35.1 | 1.2 | 44.2 |

## S2.2 On a different detection model.

In Tables S5, S6, S7 and S8, we include additional adaptation results on the PVRCNN [1] model. We use the same hyperparameters as those in the main paper. Since PVRCNN does not have the point-proposal module as in PointRCNN, we apply only PO-F and / or FB-F for adaptation. We observe our method is consistently better than baseline methods.

Table S5: **Detection performance of KITTI → Lyft adaptation with PVRCNN model.** Evaluated under $AP_{BEV}$ with IoU 0.7 for Car, 0.5 for Pedestrian and Cyclist. Please refer to Table 1 for naming.

| Method | Car | | | | Pedestrian | | | | Cyclist | | | |
|---|---|---|---|---|---|---|---|---|---|---|---|---|
| | 0-30 | 30-50 | 50-80 | 0-80 | 0-30 | 30-50 | 50-80 | 0-80 | 0-30 | 30-50 | 50-80 | 0-80 |
| No Adaptation | 61.6 | 33.9 | 10.3 | 36.9 | 29.3 | 18.1 | 0.5 | 15.2 | 34.1 | **4.5** | **0.1** | **18.2** |
| ST3D (R10)* | 62.9 | 51.5 | 27.9 | 49.1 | 20.6 | 5.6 | 0.1 | 7.0 | 31.3 | 1.9 | 0.0 | 15.0 |
| ST3D (R30) | 57.8 | 47.1 | 19.1 | 43.2 | 1.5 | 0.9 | 0.2 | 0.7 | 18.7 | 0.5 | 0.0 | 8.7 |
| PO-F (R10) | 76.4 | 62.3 | 26.7 | 60.0 | 34.9 | 23.8 | 1.8 | 17.7 | **52.4** | 0.3 | 0.0 | 9.2 |
| PO-F + FB-F (R10) | **79.7** | **67.3** | **31.9** | **64.7** | **40.4** | **30.7** | **3.7** | **23.2** | 48.1 | 0.4 | 0.0 | 10.7 |
| SN | 79.8 | 55.5 | 20.6 | 54.7 | 33.6 | 18.9 | 0.5 | 17.1 | 40.9 | 6.2 | 0.0 | 21.5 |

Table S6: **Detection performance of KITTI → Lyft adaptation with PVRCNN model.** Evaluated under $AP_{3D}$ with IoU 0.7 for Car, 0.5 for Pedestrian and Cyclist. Please refer to Table 1 for naming.

| Method | Car | | | | Pedestrian | | | | Cyclist | | | |
|---|---|---|---|---|---|---|---|---|---|---|---|---|
| | 0-30 | 30-50 | 50-80 | 0-80 | 0-30 | 30-50 | 50-80 | 0-80 | 0-30 | 30-50 | 50-80 | 0-80 |
| No Adaptation | 25.5 | 6.5 | 0.9 | 11.9 | 18.2 | 5.8 | 0.0 | 7.1 | 23.4 | **2.8** | 0.0 | **12.0** |
| ST3D (R10)* | 20.5 | 9.1 | 1.7 | 10.9 | 10.4 | 2.1 | 0.0 | 3.2 | 18.9 | 1.0 | 0.0 | 9.2 |
| ST3D (R30) | 17.3 | 9.5 | 1.7 | 9.8 | 0.6 | 0.2 | 0.0 | 0.2 | 14.9 | 0.5 | 0.0 | 7.7 |
| PO-F (R10) | 52.1 | 37.7 | 10.3 | 36.5 | **23.1** | 14.1 | 1.7 | 11.2 | 38.8 | 0.2 | 0.0 | 6.7 |
| PO-F + FB-F (R10) | **56.3** | **40.3** | **11.2** | **40.7** | 22.9 | **17.9** | **3.2** | **13.2** | **39.4** | 0.2 | 0.0 | 8.4 |
| SN | 58.1 | 21.1 | 3.5 | 28.7 | 21.6 | 9.8 | 0.2 | 9.9 | 29.5 | 3.4 | 0.0 | 15.0 |

Table S7: **Detection performance of KITTI → Lyft adaptation with PVRCNN model.** Evaluated under $AP_{BEV}$ with IoU 0.5 for Car, 0.25 for Pedestrian and Cyclist. Please refer to Table 1 for naming.

| Method | Car | | | | Pedestrian | | | | Cyclist | | | |
|---|---|---|---|---|---|---|---|---|---|---|---|---|
| | 0-30 | 30-50 | 50-80 | 0-80 | 0-30 | 30-50 | 50-80 | 0-80 | 0-30 | 30-50 | 50-80 | 0-80 |
| No Adaptation | 83.9 | 61.0 | 24.6 | 58.2 | 45.0 | 23.1 | 1.0 | 22.1 | **68.6** | **9.6** | **0.2** | **36.7** |
| ST3D (R10)* | 82.9 | 62.5 | 36.4 | 62.3 | 28.4 | 7.3 | 0.1 | 9.6 | 46.3 | 3.1 | 0.0 | 21.9 |
| ST3D (R30) | 71.3 | 54.8 | 22.7 | 51.3 | 2.5 | 1.5 | 0.5 | 1.3 | 21.8 | 0.8 | 0.0 | 9.1 |
| PO-F (R10) | 82.7 | 67.6 | 35.8 | 67.7 | 44.1 | 29.5 | 2.5 | 22.0 | 57.9 | 1.7 | 0.0 | 11.2 |
| PO-F + FB-F (R10) | **85.3** | **72.2** | **39.6** | **70.4** | **48.0** | **36.4** | **4.6** | **27.5** | 51.5 | 1.6 | 0.0 | 11.9 |
| SN | 83.0 | 61.1 | 27.3 | 59.8 | 48.0 | 24.4 | 0.8 | 23.8 | 64.5 | 10.0 | 0.2 | 35.0 |

Table S8: **Detection performance of KITTI → Lyft adaptation with PVRCNN model.** Evaluated under AP$_{3D}$ with IoU 0.5 for Car, 0.25 for Pedestrian and Cyclist. Please refer to Table 1 for naming.

| Method | Car | | | | Pedestrian | | | | Cyclist | | | |
|---|---|---|---|---|---|---|---|---|---|---|---|---|
| | 0-30 | 30-50 | 50-80 | 0-80 | 0-30 | 30-50 | 50-80 | 0-80 | 0-30 | 30-50 | 50-80 | 0-80 |
| No Adaptation | 76.9 | 47.1 | 13.4 | 47.9 | 44.8 | 22.8 | 1.0 | 22.0 | **68.4** | 8.8 | **0.2** | **36.3** |
| ST3D (R10)* | 77.3 | 54.8 | 24.8 | 54.3 | 28.3 | 7.2 | 0.1 | 9.5 | 46.2 | 3.1 | 0.0 | 21.9 |
| ST3D (R30) | 67.7 | 49.3 | 17.1 | 46.2 | 2.5 | 1.4 | 0.4 | 1.1 | 21.8 | 0.8 | 0.0 | 9.1 |
| PO-F (R10) | 82.4 | 65.6 | 31.7 | 65.3 | 44.1 | 29.5 | 2.5 | 22.0 | 57.9 | 1.4 | 0.0 | 10.9 |
| PO-F + FB-F (R10) | **85.0** | **70.2** | **36.5** | **69.4** | **48.0** | **36.3** | **4.6** | **27.4** | 51.5 | 1.1 | 0.0 | 11.6 |
| SN | 80.8 | 56.2 | 20.2 | 55.3 | 48.0 | 24.2 | 0.8 | 23.8 | 64.4 | 9.8 | 0.1 | 34.9 |

Table S9: **Detection performance of WOD → Ithaca-365 adaptation.** We evaluate the mAP as described in section 4 by different depth ranges and object types. Please refer to Table 1 for namings.

| Method | Car | | | | Pedestrian | | | |
|---|---|---|---|---|---|---|---|---|
| | 0-30 | 30-50 | 50-80 | 0-80 | 0-30 | 30-50 | 50-80 | 0-80 |
| No Adaptation | 55.7 | 38.1 | 11.7 | 37.0 | 53.0 | 33.6 | 2.1 | 32.9 |
| ST3D (R10) | 64.0 | 44.2 | 16.2 | 43.4 | 46.8 | 27.5 | 1.1 | 27.6 |
| ST3D (R30) | **67.1** | **44.7** | 17.4 | **44.2** | 39.1 | 22.1 | 1.8 | 22.3 |
| Rote-DA (Ours) | 62.5 | 44.4 | **18.9** | 43.2 | **59.9** | **42.2** | **2.8** | **35.0** |
| In Domain | 70.5 | 46.8 | 22.2 | 48.4 | 53.2 | 26.0 | 1.7 | 29.4 |

### S2.3    Additional adaptation scenario.

In Table S9 we show adaptation results for different adaptation methods adapting a PointRCNN detector trained on the Waymo Open Dataset [2] to the Ithaca-365 dataset. We observe that our conclusion holds in this case as well, especially in the class of pedestrians with a marked improvement over direct adaptation of the source model.

## S3    Additional Qualitative Visualization

Similar to Figure 4, in Figure S1 we show extra qualitative visualization of the adaptation results of various adaptation strategies in both Lyft and Ithaca-365 datasets.

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

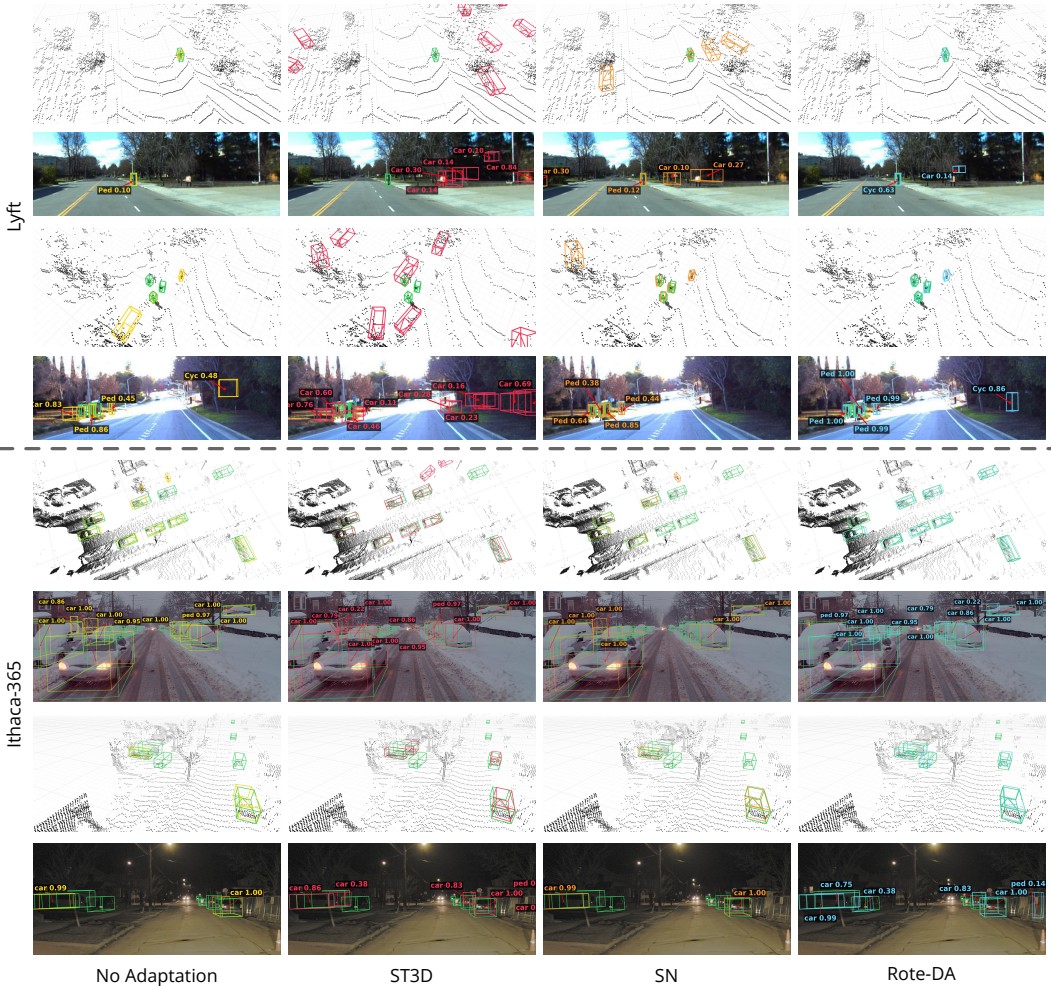

Figure S1: **Qualitative visualization of adaptation results.** We visualize two more example scenes in the Lyft and Ithaca-365 test split. Please refer to Figure 4 for more details.