# OpenReview forum: "Unsupervised Adaptation  from Repeated Traversals for Autonomous Driving"
_NeurIPS.cc/2022/Conference — NeurIPS 2022 Accept_

### Official Review · Reviewer_UWLf · 2022-07-12

**Rating:** 8
**Confidence:** 3
**Soundness:** 4 excellent
**Presentation:** 4 excellent
**Contribution:** 3 good

**Summary:**

This paper focuses on the task of unsupervised domain adaptation (UDA) for object
detection in 3D point clouds by automatically inferring additional labels in the
otherwise unlabeled target domain.

The method works best with two-stage detectors, in which case it can be applied
without modifying the model. The method is somewhat orthogonal to other UDA
techniques such as Waymo's SPG or any of the simulator-based domain adaptation
techniques, though evaluating such combinations is beyond the scope of the
paper.

The goal is to adapt a detector trained on a source environment to a new target
environment where labels are unavailable. This is done by leveraging information
from multiple passes covering the target domain. Specifically, the LiDAR
information from these passes, assumed to be co-registered, is used to compute
statistics about the likelihood that a given point belongs to a dynamic
foreground object.

These point-wise statistics are used in two ways:
 - (1) to refine the detector outputs in order to produce pseudo-labels by
   refining its predictions to remove false positives;
 - (2) as weak supervision for the proposal stage of the detector;

This process can be iterated multiple times. Predict, "clean up" predictions to
improve pseudo-labels, re-train, repeat.

The evaluation is thorough, with experiments demonstrating the generalization of
a Point-RCNN detector trained on KITTI, showcasing promising results on both the
Lyft, as well as the Ithaca-365 datasets, outperforming a couple of recent
baseline UDA methods.


**Questions:**

- [Q1] How much does PO-F still help when FB-F and FB-S are enabled? (The
  one missing line from Table 3, basically.)
- [Q2] Is the PO-F cap applied per-frame or per-dataset? I.e., do you do the
  rank-and-select within each scene, or for the whole target dataset? (My
  understanding is that you do the latter but I just wanted to double-check.)

**Limitations:**

Please see "Weaknesses" and "Questions".


**Strengths And Weaknesses:**

## Strengths
- [S1] The paper is well-written and clear, and its topic is highly relevant to
  the autonomous driving industry.
- [S2] Detailed analysis and promising results in UDA, outperforming several baselines.
- [S3] The fact that the authors plan to release the code is very helpful and would
  be a great contribution to the community.
- [S4] The labels produced by this approach can be post-processed into tracklets and
  even used to train more complex tasks such as multi-sweep detection,
  online tracking, and motion forecasting. As a suggestion, it would actually be
  nice to mention this as further possible applications of the paper, as this is
  a very important benefit of a label-oriented domain adaptation technique.

## Weaknesses
- [W1] *Single-stage vs. two-stage*: I would make it a bit clearer that a
  significant portion of the method's benefits come from the FB-S, which, in
  turn, relies on the underlying detector being two-stage. I don't doubt that
  this method can also be applied to single-stage detectors(*), but it would
  clearly require non-trivial modifications to work with SSDs.
  - (*) SSDs are still preferred in many applications because of their low
    latency, for example.
- [W2] (Minor) The improvement is only demonstrated on a single, relatively old
  LiDAR object detector (Point R-CNN). I don't think it's a dealbreaker but if I
  were to pick one thing to strengthen the paper it would be evaluating on more
  than one detector; I know OpenPCDet includes several newer detectors like
  PV-RCNN++, so it would be nice to see how those work in the proposed
  framework.

## Suggestions
- In the "Repeated Traversals" related work section, one "oldschool" area
  which has been using multiple passes for a really long time is mapping. For
  instance [levinson2010robust] use multiple passes to build a LiDAR
  intensity map, which they represent probabilistically by modeling the
  intensity of each map cell as a Gaussian. The reflectivity variance thus
  acts as a measure of the reliability of each cell for localization. While
  different from the task presented in this paper, it would be nice to mention
  this classic line of robotics work.
- L191: "way more numerous" or just "more numerous"?
- L315: "filtering" is repeated
- Citation [16] Has incorrect capitalization on one of the names.

## References
- [levinson2010robust]: Levinson, Jesse, and Sebastian Thrun. "Robust vehicle
  localization in urban environments using probabilistic maps." 2010 IEEE
  international conference on robotics and automation. IEEE, 2010.

---

> ### Author Response · Authors · 2022-08-02
> **Response to Reviewer UWLf**
>
> Thanks for your support and constructive comments! Thank you especially for the suggestion to include that our work is only the first step, and our method can be generalized into tracklets and even used to train more complex tasks. We list our response to your questions below:
> 1. Single-stage vs two stage and PVRCNN++ - Yes, we agree that the proposed FB-S currently can only be applied to two-stage models that propose boxes from point features, but it can be applicable to other models with some modifications (For example, if any voxel and its corresponding anchors cover a point, then FB-S can be applied to that voxel). Though current implementation of FB-S is specific to pointrcnn, PO-F and FB-F can still be applicable to generic detection models (Please refer to PVRCNN results below. This is the same table in our response to reviewer FkR5).
> | IoU 0.7 AP_BEV / AP_3D |     Car     |             |             |             |  Pedestrian |             |           |             |   Cyclist   |           |           |             |
> |------------------------|:-----------:|:-----------:|:-----------:|:-----------:|:-----------:|:-----------:|:---------:|:-----------:|:-----------:|:---------:|:---------:|:-----------:|
> | Model                  |     0-30    |    30-50    |    50-80    |     0-80    |     0-30    |    30-50    |   50-80   |     0-80    |     0-30    |   30-50   |   50-80   |     0-80    |
> | Direct Adapt.          | 61.6 / 25.5 | 33.9 / 6.5  | 10.3 / 0.9  | 36.9 / 11.9 | 29.3 / 18.2 | 18.1 / 5.8  | 0.5 / 0.0 | 15.2 / 7.1  | 34.1 / 23.4 | 4.5 / 2.8 | 0.1 / 0.0 | 18.2 / 12.0 |
> | SN                     | 79.8 / 58.1 | 55.5 / 21.1 | 20.6 / 3.5  | 54.7 / 28.7 | 33.6 / 21.6 | 18.9 / 9.8  | 0.5 / 0.2 | 17.1 / 9.9  | 40.9 / 29.5 | 6.2 / 3.4 | 0.0 / 0.0 | 21.5 / 15.0 |
> | ST3D (R10)             | 62.9 / 20.5 | 51.5 / 9.1  | 27.9 / 1.7  | 49.1 / 10.9 | 20.6 / 10.4 | 5.6 / 2.1   | 0.1 / 0.0 | 7.0 / 3.2   | 31.3 / 18.9 | 1.9 / 1.0 | 0.0 / 0.0 | 15.0 / 9.2  |
> | ST3D (R30)             | 57.8 / 17.3 | 47.1 / 9.5  | 19.1 / 1.7  | 43.2 / 9.8  | 1.5 / 0.6   | 0.9 / 0.2   | 0.2 / 0.0 | 0.7 / 0.2   | 18.7 / 14.9 | 0.5 / 0.5 | 0.0 / 0.0 | 8.7 / 7.7   |
> |                        |             |             |             |             |             |             |           |             |             |           |           |             |
> | PO-F (R10)             | 76.4 / 52.1 | 62.3 / 37.7 | 26.7 / 10.3 | 60.0 / 36.5 | 34.9 / 23.1 | 23.8 / 14.1 | 1.8 / 1.7 | 17.7 / 11.2 | 52.4 / 38.8 | 0.3 / 0.2 | 0.0 / 0.0 |  9.2 / 6.7  |
> | PO-F + FB-F (R10)      | 79.7 / 56.3 | 67.3 / 40.3 | 31.9 / 11.2 | 64.7 / 40.7 | 40.4 / 22.9 | 30.7 / 17.9 | 3.7 / 3.2 | 23.2 / 13.2 | 48.1 / 39.4 | 0.4 / 0.2 | 0.0 / 0.0 |  10.7 / 8.4 |
> 2. Ablation with no PO-F and with FB-F, FB-S turned on. Thank you for bringing this to our attention. We’ve included it in the table below (first row) which extends the ablations table from the main paper:
> |      |      |      |  Car |       |       |      |   | Pedestrian |       |       |      |   | Cyclist |       |       |      |
> |------|------|------|:----:|:-----:|:-----:|:----:|:-:|:----------:|:-----:|:-----:|:----:|:-:|:-------:|:-----:|:-----:|:----:|
> |      |      |      | 0-30 | 30-50 | 50-80 | 0-80 |   |    0-30    | 30-50 | 50-80 | 0-80 |   |   0-30  | 30-50 | 50-80 | 0-80 |
> | PO-F | FB-F | FB-S |      |       |       |      |   |            |       |       |      |   |         |       |       |      |
> |      |   x  |   x  | 61.2 |  40.1 |  14.6 | 40.9 |   |    41.4    |  29.7 |  0.8  | 23.4 |   |   47.5  |  2.7  |  0.0  | 23.3 |
> |   x  |      |      | 68.8 | 52.7  | 12.3  | 46.0 |   | 46.3       | 30.9  | 0.0   | 22.5 |   | 66.3    | 6.3   | 0.0   | 35.2 |
> |   x  |      |   x  | 68.4 | 55.4  | 19.6  | 49.0 |   | 41.7       | 30.8  | 1.4   | 21.8 |   | 43.3    | 1.7   | 0.0   | 21.2 |
> |   x  |   x  |      | 71.6 | 54.9  | 19.1  | 50.4 |   | 52.0       | 38.1  | 4.2   | 29.4 |   | 58.3    | 19.6  | 0.0   | 34.8 |
> |   x  |   x  |   x  | 69.0 | 58.8  | 22.6  | 52.1 |   | 48.1       | 40.8  | 2.6   | 28.7 |   | 64.7    | 26.4  | 0.0   | 40.0 |
>
> 3. PO-F cap: We rank-and-select for the whole dataset

---

> > ### Comment · Reviewer_UWLf · 2022-08-08
> > **Thank you!**
> >
> > Thank you for the detailed follow-up. The results on PV-RCNN are promising and help further cement the main claims of the paper. It's also useful to see PO-F on its own!

---

> > > ### Author Response · Authors · 2022-08-09
> > > **Thank you!**
> > >
> > > Thank you very much for your recognition and constructive discussion! We will definitely include these suggestions and improvements to refine our work for its final version.

---

### Official Review · Reviewer_YZhW · 2022-07-13

**Rating:** 5
**Confidence:** 3
**Soundness:** 3 good
**Presentation:** 3 good
**Contribution:** 3 good

**Summary:**

The paper performs iterative self-training of 3D object detectors on the target domain. Specifically, the authors assume that multiple traversals are particularly suited for end-user domain adaptation, and leverage persistency prior (PP-score) to provide a powerful signal to both false positives and false negatives from detector outputs. In this way, the detector can generate the filtered pseudo labels. This paper also introduces a new auxiliary loss that forces the detector to classify non-persistent LiDAR points as foreground.

**Questions:**

1. In Table S4 in Supplementary, why Rote-DA cannot perform well in class Car with IoU 0.5.
2. How to choose suitable hyper-parameters?

**Limitations:**

This paper proposes effective methods (i.e., posterior filtering, foreground background filtering, and foreground background supervision) to solve end-user domain adaptation with unlabeled data. There are a few limitations that this method cannot perform well in class Car at IoU 0.7/0.5 on KITTI -> Lyft adaptation, and has several hyper-parameters needed to be tuned.

**Strengths And Weaknesses:**

Strengths:
1. This paper solves end-user domain adaptation by the self-training car setting that gives rise to a weak supervision signal that is exceptionally well-suited to adapting 3D object detectors to a new environment.
2. The paper leverages unlabeled LiDAR data that automatically comes in the form of multiple traversals of the same routes to iteratively refine a detector to new domains.
3. The authors propose Posterior Filtering, Foreground Background Filtering filtering, and Foreground Background Supervision to correct and reduce both false positives and false negatives.

Weaknesses:
1. It seems that Rote-DA cannot perform well in class Car at IoU 0.7/0.5 on KITTI -> Lyft adaptation.
2. There are many hyper-parameters, e.g., alpha, beta.

---

> ### Author Response · Authors · 2022-08-02
> **Response to Reviewer YZhW**
>
> Thanks for your constructive comments! We are very happy that you agree with our simple insight, that our method is a first step towards leveraging the “weak supervision signal that is exceptionally well-suited to adapting 3D object detectors to a new environment”. We list our response to your questions below.
> 1. Rote-DA cannot perform well for Car with IoU 0.5 - Compared to direct transfer, Rote-DA is overall better (though slightly worse in the 0-30 range but the improvements for the other two ranges is significant). As for comparison to ST3D, we suspect the use of the memory bank of previous high quality predictions is helping the model’s recall (thus better performance at low IOU)
> 2. Choosing hyperparameter - choosing hyperparameter is an active research question on learning from unlabeled data (e.g semi-supervised learning, self-supervised learning, unsupervised domain adaptation etc). We selected the best hyperparameters based on the performance on lyft and used the same hyperparameter for KITTI → Ithaca365. This shows that the set of hyperparameters we selected based on lyft is fairly robust and generalizable. We realized we did not make this clear in the main text, and we will modify the main text accordingly.

---

> ### Author Response · Authors · 2022-08-09
> **Please let us know if you have additional concerns or questions**
>
> Hello dear reviewer, we just wanted to ask if you have any additional questions or concerns? Just a gentle reminder that today is the last day for discussion, and we were wondering if there is any other information we can provide to help you with your review. Thanks in advance!

---

### Official Review · Reviewer_mdTS · 2022-07-13

**Rating:** 3
**Confidence:** 4
**Soundness:** 2 fair
**Presentation:** 3 good
**Contribution:** 2 fair

**Summary:**

The paper tackles the task of 3D object detection in LiDAR point clouds, particularly detecting moving objects such as cars, pedestrians, and cyclists, with a strong emphasis on applications for autonomous ground vehicles. The authors leverage past traversals from the target domain, in order to refine pseudo-labels (using 3 refinement procedures: PO-F, FB-F, and FB-S) and adapt a pre-trained (on the source domain) PointRCNN model in a bid to improve generalization without the effort of novel annotations (by refining predictions on unlabeled data).

**Questions:**

* How does your work compare to Hindsight (reference [36] in the paper)? Based on the reported numbers on Lyft they perform better, but I did not check carefully if the evaluation scenario is similar. Is it feasible to do a fair numerical comparison between your method and theirs?

* Regarding statement from L93-94 - Do the authors have any experiments to back up this theory? How do the authors know the gains would be marginal? Numerical arguments or at least a more thorough argument.

* Regarding the statement from L224-226 - What about the nuScenes dataset [1]? Why wasn't it included in the list? Hindsight uses past traversals and reports numbers on nuScenes.
[1] - Caesar, H., Bankiti, V., Lang, A. H., Vora, S., Liong, V. E., Xu, Q., ... & Beijbom, O. (2020). nuscenes: A multimodal dataset for autonomous driving. In Proceedings of the IEEE/CVF conference on computer vision and pattern recognition (pp. 11621-11631).

* The numbers using ST3D look terrible, especially for R30. The authors should state in the paper if this is a reimplementation and replicate one of the experiments reported in the ST3D paper to make results credible (which could be an easy task since ST3D uses 4 datasets: Waymo, KITTI, Lyft, and nuScenes). And again my question regarding the limitation to using just KITTI as a source domain? At least something comparable to what is presented in ST3D - "(1) adaptation from label rich domains to label insufficient domains (i.e., Waymo to other datasets) and (2) across domains with different numbers of the LiDAR beams (i.e., Waymo → nuScenes and nuScenes → KITTI)"

**Limitations:**

Privacy concerns properly addressed in the "Conclusion" section, but I did not find a dedicated section addressing the current limitations of the proposed method (I think this could be very useful). Some of the initial assumptions could be considered limitations (e.g. (1) the fact that this method works only if the objects of interest are dynamic, (2) the unlabeled data from the target domain are collected from the same routes at a relatively close period of time and (3) the source and target domains are not from drastically different areas (city vs. farmland)).

**Strengths And Weaknesses:**

Strengths:
* Originality: similar work has been done before, but I am not aware of any pseudo-label refinement procedures in LiDAR point clouds using the persistency scores
* Quality: the paper is well-written and well-structured, all details needed for reproducibility have been provided
* Clarity: Good to see the full thought process for each of the steps involved in the method (backed up by ablative studies - Sec. 4.2) although there are some aspects in the evaluation procedure that create confusion - e.g. L242-243 vs. L244-245 conflicting evaluation procedures? Is it a frontal view or BEV evaluation?

Weaknesses:
* Significance: Some arguments are not backed up by proper experimental analysis - e.g. L184 - the values for those thresholds are hard to swallow without proper analysis - these were validated on which dataset? Do they apply for both Lyft and Ithaca or is validation done on KITTI? Also, the experimental setup is limited to only one source domain (KITTI) compared to previous work e.g ST3D which varied the source and target domains.

Minor comments:
* Figure 1 is placed on page 3 and it's first presented on page 5 - maybe it should be placed on the page closer to the text reference
* L103 - Remove "to" - "... uses the signal to _to_ learn"
* L111-112 - complementary work to very recently published work which is a plus
* Table 3 caption - use "two" instead of "to" in "... can be further subdivided into _two_ subcomponents"
* Table 3 - which testing scenario did you report the numbers on - failed to find this information in the text as well.
* L191 - rephrase - remove first "is" or the second, but keeping both makes no sense in "... One naive way is to remove them _is_ by thresholding"

---

> ### Author Response · Authors · 2022-08-02
> **Response to Reviewer mdTS (Part 1)**
>
> Thank you for the positive opinion of our work, and for finding it original, of high quality, and clearly explained/studied! We list our response to your questions/concerns below. We truly value the points you raise. However, we are a little surprised by the negative final recommendation (reject) and are not entirely sure which of your points gave rise to it. However, as we are confident that we can address all of your concerns we are hopeful that you will be able to raise your score after the rebuttal period.
>
> 1. Evaluation (BEV vs frontal) — We restrict the scenes to frontal view and report the performance from either BEV or 3D view of the restricted scene. Similar evaluation scheme has been adopted in various papers [1, 2]. Nonetheless, we realize that our wording could be confusing, and we will modify the main text for better clarity. Thanks for the suggestion!
> 2. L184, hyperparameter selection: We selected the best hyperparameters based on the performance on KITTI → lyft and used the same hyperparameter for KITTI → Ithaca365. This shows that the set of hyperparameters we selected based on lyft is fairly robust and generalizable. We realized we did not make this clear in the main text. We will modify the text accordingly.
> 3. Multiple sources - Our method does work across multiple sources; thank you for this suggestion, and we will include experiments to show this! We include results for our method trained on the Waymo dataset adapted for Ithaca365 dataset below:
> | nuSc. mAP     |  Car |       |       |      |   | Pedestrian |       |       |      |
> |---------------|:----:|:-----:|:-----:|:----:|---|:----------:|:-----:|:-----:|:----:|
> | Method        | 0-30 | 30-50 | 50-80 | 0-80 |   |    0-30    | 30-50 | 50-80 | 0-80 |
> | Direct Adapt. | 55.7 |  38.1 |  11.7 | 37.0 |   |    53.0    |  33.6 |  2.1  | 32.9 |
> | ST3D (R10)    | 64.0 |  44.2 |  16.2 | 43.4 |   |    46.8    |  27.5 |  1.1  | 27.6 |
> | ST3D (R30)    | 67.1 |  44.7 |  17.4 | 44.2 |   |    39.1    |  22.1 |  1.8  | 22.3 |
> | Ours (R10)    | 62.5 |  44.4 |  18.9 | 43.2 |   |    59.9    |  42.2 |  2.8  | 35.0 |
> | Reference     | 70.5 |  46.8 |  22.2 | 48.4 |   |    53.2    |  26.0 |  1.7  | 29.4 |
>
> Observe that our conclusion holds in this case as well, especially in the class of pedestrians with a marked improvement over direct adaptation of the source model. We will include this in the final version of the paper.
>
> 4. Comparison to Hindsight - The results are not actually directly comparable to Hindsight, since Hindsight uses repeated traversals also in test time (we use repeated traversals for adaptation but do not assume access to repeated traversals when testing the adapted model). However, our method can be orthogonal to Hindsight if we assume that the source model is a Hindsight model (i.e. repeated traversals at the source domain) and we have repeated traversals during test-time after adaptation.
>
> 5. Line 93-94 - We make such a statement because Rote-DA’s performance of Pedestrian and Cylist is almost reaching the same performance as the in-domain detector (for KITTI to Lyft). However, we do agree that combining different methods could potentially bring forth improvements to other classes or when transferring between different source and target datasets so we will reword this section in the final version
>
> 6. For nuScenes: the localization is actually not quite accurate on z (https://www.nuscenes.org/nuscenes#data-format) (the z offset is always zero according to the documentation)
>
> 7. ST3D - The official codebase for ST3D only works for single class detection (i.e car detection) so we made minimal modifications to their codebase — similar to a [Github issue](https://github.com/CVMI-Lab/ST3D/issues/49#issuecomment-1197591322) which have shown success. Also, our small modifications do in fact bring forth improvements for car detection (for all setup, as validated in the original paper), and the pedestrian class for KITTI → Ithaca365. As for the poor performance of the non-car classes in KITTI → Lyft, we want to point out the fact that the performance of ST3D could be unstable, as evidenced by a [Github issue](https://github.com/CVMI-Lab/ST3D/issues/30) and by the authors themselves (the authors suggested looking at the “best” model [here](https://github.com/CVMI-Lab/ST3D/blob/master/docs/GETTING_STARTED.md). We did not resort to picking the best model since it is not realistic to assume labeled target data. As a compromise, we decided to report the results at epoch 30 (which is the default training epoch) and at epoch 10 (which is the number of epoch for our approach) to show (a) ST3D is unstable and (b) with the same amount of training epoch, ROTE-DA is much better than ST3D).
> (to be continued...)

---

> > ### Author Response · Authors · 2022-08-02
> > **Response to Reviewer mdTS (Part 2)**
> >
> > 8. Limitations - Duly noted and we will update the text to include limitations. Below are our discussion on some of the limitations
> >       - Dynamic Objects Only and how to adapt to static objects - FB-F and PO-F could be adapted for static objects easily (for FB-F, we could remove static object predicted boxes that have too many ephemeral points). Though FB-S could not be used for static objects, one could imagine adapting the source detectors into two - one for dynamic objects (with FB-S to reap the benefits of our approach) and another one for static, and applying NMS to remove overlapped boxes from the two detectors. Of course, this would be a study left for future work and is a limitation of our current method.
> >       - Unlabeled data collected from the same routes at a relatively close period of time – this is indeed our assumptions/limitations. One argument for this assumption is that this is actually a realistic scenario since a lot of us do drive through the same route again and again for work and school in a close period of time (e.g every week day)
> >       - No drastic difference in source and target domain: Indeed, we made an assumption that the target and source domain share the same class distribution for PO-F but this is realistic since with modern GPS, we do know whether the data is collected from a city or farmland (and we can also tune PO-F based on the location). Besides, we purposely select source and target datasets that are collected in different parts of the world with potentially different object sizes and road users behavior (e.g Cars in Germany (KITTI) have different sizes compared to cars in the USA [2]) and weather conditions (e.g KITTI is collected in sunny conditions whereas Ithaca365 covers various weather conditions). Thus, there are big domain gaps between the source and target domain we selected as evidenced by the performance gap between the in-domain detector and directly applying the source domain detector in table 1 and 2.
> >
> > [1] Andreas Geiger, Philip Lenz, Christoph Stiller, and Raquel Urtasun. Vision meets robotics: The 348 kitti dataset. The International Journal of Robotics Research, 32(11):1231–1237, 2013
> >
> > [2]  Yan Wang, Xiangyu Chen, Yurong You, Li Erran Li, Bharath Hariharan, Mark Campbell, Kilian Q. Weinberger, and Wei-Lun Chao. Train in germany, test in the usa: Making 3d object detectors generalize. In CVPR, pages 11713–11723, June 2020

---

> ### Author Response · Authors · 2022-08-09
> **Please let us know if you have additional concerns or questions**
>
> Hello dear reviewer, just a gentle reminder that today is the last day for discussion. We have provided more results and explanations based on your concerns. Could you kindly go over them and let us know whether you have additional questions or not? Thanks in advance!

---

### Official Review · Reviewer_FkR5 · 2022-07-13

**Rating:** 5
**Confidence:** 4
**Soundness:** 2 fair
**Presentation:** 3 good
**Contribution:** 3 good

**Summary:**

This work proposes Rote-DA, a method for unsupervised domain adaptation for 3D detectors.
By using LIDAR readings over multiple traversals in the target domain,
they use the "persistence prior score" (PP-score, prior work) to filter out false-positives and provide additional supervision for reducing false negatives.
They conduct experiments on the Ithica-365 and Lyft datasets and show their adaptation strategy improves adapted detector performance.

**Questions:**

* Is there any easy way to adapt this method to handle road divider / traffic cones / street lights / signs?
* It seems the self-training only improves the cyclist class - can the authors give any insight on this?
* Have there been any experiments with other architectures? Mostly curious about single-stage detectors.
* How many traversals of the same scene are in the datasets?

### Miscellaneous
* Technical portion is well written but a figure summarizing the filtering would be nice to look at
* The method is motivated by the fact that repeated traversals will be collected by end users, but for now most (if not all) cars do not have lidar sensors.
* FYI - the nuScenes dataset has repeated traversals

**Limitations:**

yes

**Strengths And Weaknesses:**

### Strengths
* Written clearly, technical portion straightforward and well inspired
* Simple and improved performance

### Concerns
* Only provides comparisons to two other methods
* Method only shown on PointRCNN
* Slightly worried about technical novelty - FB-F and FB-S could be seen as just a few filtering tricks
* Applicable to only dynamic objects (doesn't work for road divider, traffic cone, street lights/signs)

---

> ### Author Response · Authors · 2022-08-02
> **Response to Reviewer FkR5 (Part 1)**
>
> Thanks for your constructive comments and for finding our paper well written and our method’s performance strong! In addition, the suggestion regarding a figure to “summarize the filtering” is duly noted and will definitely be included in the final version. We list our response to your questions below.
> 1. Limited Comparisons: Our work focuses on source-free domain adaptation, i.e., how to adapt a pre-trained detector without access to data from the source domain. This setup is extremely practical and important since source data might not always be available due to various reasons (e.g. privacy concerns, storage, etc). Relevant work in source free domain adaptation is still limited. To the best of our knowledge, the few relevant works are [1, 3, 4, 5, 6]. We picked [5, 6] as our baselines given their simplicity and because ST3D[6] uses a better variant of self-training. [3, 4]  leverages temporal information (i.e. tracking) which we do not assume access but is complementary to repeated traversals (we leave exploration on combining both sources of information for future work). As for SPG[1], we did not compare them since (a) it is catered to a specific type of domain gap (change in lidar distribution due to weather conditions) and (b) it is extremely complex without any publicly available code. But, we stress that SPG is orthogonal to our approach, and can in principle be combined for better performance.
> 2. Only using PointRCNN and Experimentation with Other Architectures - Thanks for the suggestion! In fact, our method is not limited to two-stage and point-based. For instance, PO-F and FB-F should be applicable since it only looks at boxes. For FB-S, it can be applied on methods that propose boxes from point level features, but it also can be applicable to other models with some modifications (For example, if any voxel and its corresponding anchors cover a point, then FB-S can be applied to that voxel). We experiment with PVRCNN on PO-F and FB-F (results shown below). It can be seen that even with PO-F and FB-F, our method is still better than baseline methods.
> | IoU 0.7 AP_BEV / AP_3D |     Car     |             |             |             |  Pedestrian |             |           |             |   Cyclist   |           |           |             |
> |------------------------|:-----------:|:-----------:|:-----------:|:-----------:|:-----------:|:-----------:|:---------:|:-----------:|:-----------:|:---------:|:---------:|:-----------:|
> | Model                  |     0-30    |    30-50    |    50-80    |     0-80    |     0-30    |    30-50    |   50-80   |     0-80    |     0-30    |   30-50   |   50-80   |     0-80    |
> | Direct Adapt.          | 61.6 / 25.5 | 33.9 / 6.5  | 10.3 / 0.9  | 36.9 / 11.9 | 29.3 / 18.2 | 18.1 / 5.8  | 0.5 / 0.0 | 15.2 / 7.1  | 34.1 / 23.4 | 4.5 / 2.8 | 0.1 / 0.0 | 18.2 / 12.0 |
> | SN                     | 79.8 / 58.1 | 55.5 / 21.1 | 20.6 / 3.5  | 54.7 / 28.7 | 33.6 / 21.6 | 18.9 / 9.8  | 0.5 / 0.2 | 17.1 / 9.9  | 40.9 / 29.5 | 6.2 / 3.4 | 0.0 / 0.0 | 21.5 / 15.0 |
> | ST3D (R10)             | 62.9 / 20.5 | 51.5 / 9.1  | 27.9 / 1.7  | 49.1 / 10.9 | 20.6 / 10.4 | 5.6 / 2.1   | 0.1 / 0.0 | 7.0 / 3.2   | 31.3 / 18.9 | 1.9 / 1.0 | 0.0 / 0.0 | 15.0 / 9.2  |
> | ST3D (R30)             | 57.8 / 17.3 | 47.1 / 9.5  | 19.1 / 1.7  | 43.2 / 9.8  | 1.5 / 0.6   | 0.9 / 0.2   | 0.2 / 0.0 | 0.7 / 0.2   | 18.7 / 14.9 | 0.5 / 0.5 | 0.0 / 0.0 | 8.7 / 7.7   |
> |                        |             |             |             |             |             |             |           |             |             |           |           |             |
> | PO-F (R10)             | 76.4 / 52.1 | 62.3 / 37.7 | 26.7 / 10.3 | 60.0 / 36.5 | 34.9 / 23.1 | 23.8 / 14.1 | 1.8 / 1.7 | 17.7 / 11.2 | 52.4 / 38.8 | 0.3 / 0.2 | 0.0 / 0.0 |  9.2 / 6.7  |
> | PO-F + FB-F (R10)      | 79.7 / 56.3 | 67.3 / 40.3 | 31.9 / 11.2 | 64.7 / 40.7 | 40.4 / 22.9 | 30.7 / 17.9 | 3.7 / 3.2 | 23.2 / 13.2 | 48.1 / 39.4 | 0.4 / 0.2 | 0.0 / 0.0 |  10.7 / 8.4 |
> 3. Limited Technical Novelty - While FB-S and FB-F are indeed “filtering tricks”, our contribution lies in observing that repeated traversals provide the needed information for performing such filtering, and combining this filtering with detector training pipelines. The simplicity of our approach is in fact a strength; it shows that with the right signal derived from multiple traversals, state-of-the-art domain adaptation can be done with simple filtering. Simple approaches are also easier to adopt and less prone to bugs/failures.
> (to be continued....)

---

> > ### Author Response · Authors · 2022-08-02
> > **Response to Reviewer FkR5 (Part 2)**
> >
> > 4. Dynamic Objects Only and how to adapt to static objects - FB-F and PO-F could be adapted for static objects easily (for FB-F, we could remove static object predicted boxes that have too many ephemeral points). Though FB-S could not be used for static objects, one could imagine adapting the source detectors into two - one for dynamic objects (with FB-S to reap the benefits of our approach) and another one for static, and apply NMS to remove overlapped boxes from the two detectors.
> > 5. The number of traversals - We are using all available traversals provided by the dataset and the number of traversals varies from 2 to 10 for different locations.
> > 6. Self-training only improves Cyclist - Could you point us to which table you are referring to? We observe significant improvement for all classes in table 1 in the main text.
> > 7. Prevalence of LiDAR sensor - Indeed, the adoption of LiDAR is limited currently. But most of the L4-self-driving cars companies adopt a LiDAR-centric perception system. So as more self-driving cars or cars with smart assistance systems deploy, there will be more cars with LiDAR. Besides, our approach works on any point-cloud, so we can also leverage point-cloud from stereo cameras.
> > 8. For nuScenes: we actually have looked into the nuScenes dataset but we found the localization is actually not quite accurate on z (https://www.nuscenes.org/nuscenes#data-format) (the z offset is always zero according to the documentation). We thus did not choose to experiment on the nuScenes dataset. We are happy to update the results on nuScenes in the future if the localization issue is resolved.
> >
> > [1] Qiangeng Xu, Yin Zhou, Weiyue Wang, Charles R Qi, and Dragomir Anguelov. Spg: Unsupervised domain adaptation for 3d object detection via semantic point generation. In Proceedings 422 of the IEEE/CVF International Conference on Computer Vision, pages 15446–15456, 2021.
> >
> > [3] Cristiano Saltori, Stéphane Lathuiliére, Nicu Sebe, Elisa Ricci, and Fabio Galasso. Sf-uda3d: Source-free unsupervised domain adaptation for lidar-based 3d object detection. In 2020  International Conference on 3D Vision (3DV), pages 771–780. IEEE, 2020
> >
> > [4] Yurong You, Carlos Andres Diaz-Ruiz, Yan Wang, Wei-Lun Chao, Bharath Hariharan, Mark Campbell, and Kilian Q. Weinberger. Exploiting playbacks in unsupervised domain adaptation for 3d object detection. In Proceedings of the IEEE International Conference on Robotics and Automation (ICRA), May 2022.
> >
> > [5] Yan Wang, Xiangyu Chen, Yurong You, Li Erran Li, Bharath Hariharan, Mark Campbell, Kilian Q. Weinberger, and Wei-Lun Chao. Train in germany, test in the usa: Making 3d object detectors generalize. In CVPR, pages 11713–11723, June 2020
> >
> > [6] Jihan Yang, Shaoshuai Shi, Zhe Wang, Hongsheng Li, and Xiaojuan Qi. St3d: Self-training for unsupervised domain adaptation on 3d object detection. In Proceedings of the IEEE/CVF Conference on Computer Vision and Pattern Recognition, 2021.

---

> > > ### Comment · Reviewer_FkR5 · 2022-08-09
> > > **Response to Response to Reviewer FkR5**
> > >
> > > Thank you for the clarifications, please include some of these in the final revision, particularly (2, 4, 5)
> > >
> > > 2. Nice to see these additional experiments, these should definitely be included
> > > 6. Table 1, comparing against baseline SN in the full range ("0-80")

---

> > > > ### Author Response · Authors · 2022-08-09
> > > > **Thank you!**
> > > >
> > > > Thank you for your support, as well as the suggestions and helpful comments. We will definitely include these in our final version, with particular care for points 2, 4, and 5. We were a bit confused by what you mean for "Table 1, comparing against baseline SN in the full range"? We currently do have that comparison in Table 1 of the main paper. We will definitely emphasize this comparison in the final version. Let us know if there's any additional information or clarifications we can provide!

---

### Meta-Review · Area_Chair_yMiL · 2022-08-20

**Recommendation:** Accept
**Confidence:** Less certain

**Metareview:**

This paper proposes to leverage unlabelled LIDAR scans from vehicles repeatedly traversing the same environment (e.g., Lyft data in the US), for the domain adaptation of car, bicycle and pedestrian classifiers trained in different domains (e.g., KITTI data in Germany). It uses pretrained Point R-CNN classifiers and bounding box detectors followed by Persistent Point PP-scores and statistical methods for false positive and false negative removal. Various ablations demonstrate the superiority of this self-training method.

Reviewers had very split scores, going from 8 (UWLf) down to 3 (​​mdTS). After rebuttals, reviewer's estimated score is 5 (not updated in the review, but the reviewer said they were "favourable"), meaning an average score of 5.75.

Reviewers praised the idea (UWLf, YZhW, mdTS), the clarity of the paper (UWLf, FkR5, mdTS), the thorough evaluation (UWLf, mdTS), the promising generalisation results on the Lyft and Ithaca-365 datasets (UWLf, YZhW, FkR5) and the potential uses of unsupervised data labeling (UWLf).

Reviewer (UWLf, FkR5) noted that other LIDAR point classifiers other than Point R-CNN or single-stage detectors could have been evaluated: these points were addressed during rebuttals with new experiments). Reviewer (YZhW) noted poor performance in one specific domain and the presence of many hyperparameters, but did not respond to the authors’ rebuttal. Reviewer (FkR5) was concerned about applicability to static objects, limited novelty of the filtering methods, and the fact that comparison was  with only two other methods; the authors provided a rebuttal to most of these points. Reviewer mdTS had a large number of specific questions regarding the evaluation, starting with the hyperparameter choice; again, the reviewer did not answer to the authors’ rebuttal.

As AC, and based on scores 8, 5, 5, and 3->5?, I would recommend this paper for acceptance.

Sincerely,
Area Chair

**Award:**

No

---

### Decision · Program_Chairs · 2022-09-14

Accept